# Learning and Expertise in Mineral Exploration Decision-Making: An Ecological Dynamics Perspective

**DOI:** 10.3390/ijerph18189752

**Published:** 2021-09-16

**Authors:** Rhys Samuel Davies, Marianne Julia Davies, David Groves, Keith Davids, Eric Brymer, Allan Trench, John Paul Sykes, Michael Dentith

**Affiliations:** 1Business School, The University of Western Australia, Perth, WA 6009, Australia; Allan.Trench@uwa.edu.au (A.T.); john.sykes@greenfieldsresearch.com (J.P.S.); 2Centre for Sport & Exercise Science, Sheffield Hallam University, Sheffield S10 2BP, UK; marianne.j.davies@student.shu.ac.uk (M.J.D.); k.davids@shu.ac.uk (K.D.); 3Centre for Exploration Targeting, School of Earth Sciences, The University of Western Australia, Perth, WA 6009, Australia; di_groves@hotmail.com; 4Faculty of Health, Gold Coast Campus, Southern Cross University, Gold Coast, QLD 4225, Australia; eric.brymer@scu.edu.au; 5School of Earth Sciences, The University of Western Australia, Perth, WA 6009, Australia; michael.dentith@uwa.edu.au

**Keywords:** mineral exploration, ecological dynamics, expertise, needs-supportive environment, representative learning design

## Abstract

The declining discovery rate of world-class ore deposits represents a significant obstacle to future global metal supply. To counter this trend, there is a requirement for mineral exploration to be conducted in increasingly challenging, uncertain, and remote environments. Faced with such increases in task and environmental complexity, an important concern in exploratory activities are the behavioural challenges of information perception, interpretation and decision-making by geoscientists tasked with discovering the next generation of deposits. Here, we outline the Dynamics model, as a diagnostic tool for situational analysis and a guiding framework for designing working and training environments to maximise exploration performance. The Dynamics model is based on an Ecological Dynamics framework, combining Newell’s Constraints model, Self Determination Theory, and including feedback loops to define an autopoietic system. By implication of the Dynamics model, several areas are highlighted as being important for improving the quality of exploration. These include: (a) provision of needs-supportive working environments that promote appropriate degrees of effort, autonomy, creativity and technical risk-taking; (b) an understanding of the wider motivational context, particularly the influence of tradition, culture and other ‘forms of life’ that constrain behaviour; (c) relevant goal-setting in the design of corporate strategies to direct exploration activities; and (d) development of practical, representative scenario-based training interventions, providing effective learning environments, with digital media and technologies presenting decision-outcome feedback, to assist in the development of expertise in mineral exploration targeting.

## 1. Introduction

Despite increased expenditure, global greenfield discovery rates have stalled for over a decade (Figure 1 [1]). Reversal of the current trend is deemed critical for the desired global transition to renewable energy sources, particularly wind and solar, as well as the evolution of currently energy intensive industries, such as the automotive industry moving from fossil fuels to predominantly battery-driven vehicles [2]. Without access to raw materials, especially those defined as critical metals, society is unlikely to realise long-term goals towards achieving sustainability [3,4,5]. Additionally, well-targeted exploration will help reduce the environmental footprint of conducting exploration activities by decreasing the average number of holes drilled to make each discovery.

Traditionally, mineral exploration has followed a predominantly empirical approach, which initially entails searching for evidence of mineralisation upon the surface of the planet, then, with consideration of known deposit types, targeting drilling to define the extent of an ore body below the surface [6]. As outcropping deposits in well-explored areas are progressively depleted, there is a need for exploration to extend to less well-explored search spaces, typically in more remote locations, and for undefined deposit types to be considered [7]. These points suggest that exploration needs to include regions where surficial evidence for deposits is absent or unclear, requiring a conceptual approach to targeting, guided by an understanding of underlying mineralising processes shared across multiple deposit types (Mineral Systems Concept [8]). As the industry experiences this transition, Davies and Davies [9] argue that creativity in the application of the Mineral Systems Concept is key to realising long-term exploration success.

This paper presents the Dynamics model [9], incorporating the Dynamics challenge-performance curve [10] as a principled framework for understanding and supporting creativity and the development of expertise in predictive exploration targeting. These models provide guidance to the minerals industry in identifying and realising current decision constraints and adapting learning and working environments to promote greater degrees of creativity and on-going development of exploration targeting expertise.

### 1.1. Exploration Targeting

Exploration targeting involves defining and exploring areas that have potential to host economic mineralisation. This process, outlined in Figure 2, is recognised as a series of decisions [11,12,13]. In this decision-making process, both explicit and tacit (or implicit) knowledge is employed to assess the validity of various options available to the decision-maker [14]. At the time of the decision, the outcome of each option cannot be known for certain but is inferred through assessment of the information available [15]. To improve the quality of an assessment, and therefore the likelihood of a positive decision-outcome, an individual or team is required to call upon relevant experience from a host of disciplines. Specifically, exploration decision-making encompasses a range of geoscience-related disciplines [16], as well as general disciplines such as economics, business strategy, management and socio-political implications that may influence license to operate [17]. Exploration targeting thus represents a highly complex and dynamic task.

### 1.2. The Exploration Search Space

Mining activities constitute sampling without replacement, from a fixed but unknown number of deposits, within any given search space [16]. Once resources are extracted from a location, they are permanently depleted. Ongoing exploration of locations containing high data density, significant known mineralisation and previous mining activities thus presents a long-term declining rate of return [18,19].

The current decline in exploration success is likely due to known search spaces reaching maturity [20,21]. In order to reverse the declining rate in exploration success, there is a need for sustained greenfield exploration in immature or newly discovered search spaces, often in remote or extreme environments [22]. For this to occur, explorers must move away from locations of high data density to areas of greater uncertainty with limited information, but also greater opportunity for significant new mineral discoveries [23]. This approach to the process of exploration targeting carries inherent risks and challenges.

One popular new frontier is exploration for buried deposits beneath transported, post-mineral cover. When buried under recently transported material, mineralisation is often impossible to detect by surficial exploration methods. Exploration for deposits in areas with considerable cover, including much of Australia, requires a conceptual targeting approach. Conceptual targeting requires exploration geologists to utilise available information so as to predict the likelihood that an economic ore body exists within a given search space. Rather than looking for empirical evidence of mineralisation, such an approach is underpinned by a detailed understanding of the processes leading to ore deposit formation and associated physical and chemical footprints [8]. In conducting conceptual targeting, the exploration geologist requires datasets that map mineralising processes and knowledge of a wide range of related deposit types [24].

### 1.3. The Mineral Systems Concept

Ore deposit formation is recognised to be a focused mineralisation event. These events represent “self-organising” critical systems and are underpinned by the interaction of complex, non-linear deposit forming processes [25]. Based on an understanding of these processes, the Mineral Systems Concept provides a framework for conceptual exploration targeting [26]. However, the application of an improved understanding of mineralisation systems is yet to provide a new wave of exploration discoveries [11]. 

Application of the Mineral Systems Concept to define new search spaces or predict new deposit types is conceptually challenging. This approach requires the development and testing of hypotheses, based on an understanding of mineralisation processes, to predict the potential for existing and/or undefined deposit types to exist within a search space [17]. For cases where exploration is conducted in newly defined search spaces, there are often limited geoscientific datasets, creating a significant barrier to entry in that new data acquisition is an obstacle. Where data representative of system elements are available, it is difficult to integrate multiple geoscientific datasets and model complex interactions between mineralisation processes. Most importantly, the development of predictive targeting hypotheses requires significant creative input, based on a rich and diverse background of knowledge and understanding. Due to the risks and low success rates associated with these search challenges, there is a heuristic tendency for explorers to be biased towards preferential focus on advanced, data-dense ‘brownfields’ projects, over conceptual early stage ‘greenfields’ projects.

### 1.4. Technology as a Solution

Hronsky and Groves [26] recognise the propensity for mineral discoveries to follow the emergence of new concepts or technologies. As such, technological development has been proposed as part of the solution to the current decline in exploration success [27]. Mineral exploration is becoming increasingly data rich and knowledge poor, such that advanced algorithms and increasing computational power may provide opportunity to recognise patterns in larger, more complex datasets [27]. However, a critique of solely technology-orientated solutions is that they may fail to resolve scenario or context-related issues that need to be considered in the search process [14]. For example, although development of new geophysical survey methods may present improved mapping of components of mineral system processes [28], this fails to provide optimal survey locations, prior to data collection commencing, or methods for integrating newly acquired data into the broader, highly complex exploration decision-making process.

Greenfield exploration requires explorers to operate in data-poor environments [23]. They are required to fill significant gaps in existing geoscientific datasets, to predict potential decision outcomes and integrate with ongoing exploration activities [29]. By relying solely on technological development, we may fail to consider the requirement for human innovation and creative problem-solving in the generation and critique of multiple predictions, based on disparate datasets. Techniques such as ‘deep learning’ have provided significant progress in the ability of Artificial Intelligence (AI) to begin exploring a creative frontier [30]. Unfortunately, this creativity has been critiqued as merely mimicking existing data rather than producing genuinely novel outcomes, especially for challenging, high-dimensional problems [31]. With regard to the critique of predicted hypotheses, current AI is limited in its ability to interrogate correlations and avoid biases since current deep learning methods are poorly correlated with prior knowledge [31,32]. 

By providing comprehensive training data, AI appears capable of predicting locations of well-understood deposits in data-rich environments [33]. Nevertheless, AI remains far from being able to generate new conceptual deposit types or search spaces, based on a holistic understanding of Earth sciences. In an article on the subject of learning, Gopnik [34] states that “Despite enormous strides in machine intelligence, even the most powerful computers still cannot learn as well as a five-year-old does.” She identifies the care, nurturing and support provided to a child as key ingredient in learning and creativity. In summary, mineral exploration is recognised as a complex, non-linear process [26] and although advances in AI and technology present significant opportunities to improve the quality of exploration targeting, the combination of technology and human creativity are considered key to realising long-term, recurring success [10].

### 1.5. Creative Problem Solving

Creative thinking is key to the development of novel solutions to overcome complex, high-dimensional problems. Within mineral exploration, previous explorers conducted targeting to test specific hypotheses. Since mining activities involve sampling without replacement, economic discoveries realised by previous explorers no longer remain within the search space. As such, it is important that future exploration efforts test newly developed hypotheses regarding new search spaces and subtly different deposit types. 

Through consideration of underlying system processes, often shared between multiple deposit types, the Mineral Systems Concept presents a robust, scientific framework for creative problem solving. By reviewing existing empirical information, it is possible to separate features that are likely to be representative of key ore forming processes that define a mineral system, from those that are only locally relevant to a deposit [24]. Primed with this knowledge, an explorer can generate innovative, science-based hypotheses regarding new search spaces in which the same mineralisation processes have occurred or differing local features that may result in new deposit types. This approach allows the explorer to recognise fundamental patterns across geological settings and styles of mineralisation, thus generating hypotheses to test new exploration targets. However, creative problem solving and innovation are not simply academic endeavours. They require a high level of individual confidence and a perceived supportive environment [35].

### 1.6. Subjectivity in Geoscience

Subjectivity is the norm; a surprisingly small number of geoscience-specific studies have delved into the process of predictive decision-making and subjective assessment. Polson and Curtis [36] and Bond, Lunn, Shipton and Lunn [37] discuss the role of heuristics in generating geological hypotheses, or interpretations, based on uncertain data. They highlighted the potential for experts to reach contradictory conclusions when analysing the same data, advising that group workshops can help to reduce bias through sharing of knowledge and expertise [38]. Although these papers provide important insights into the elicitation process and adequacy of group assessment workflows, they fail to discuss the importance of developing expertise and the role it has in influencing the quality of interpretations. This omission is possibly due to an assumption that participants were already experts. Only Bond et al. [37] make comment that having a Masters or Ph.D. qualification significantly improved expert performance in interpreting a seismic dataset. Davies et al. [39] conducted a group workshop to evaluate the orogenic gold endowment of a greenstone belt in Western Australia, noting a significant degree of variation in expert estimates, but failing to find a relationship between estimates and participant experience. This issue presents an important question regarding the role of training and expertise in generating accurate or realistic hypotheses during creative problem-solving tasks.

### 1.7. Ecological Dynamics

Future exploration success relies on relevant skills of structured and creative perception, decision-making, and the planning of exploration targeting activities in challenging, uncertain and often remote environments. Targeting new conceptual deposit types or search spaces, based on a holistic understanding of Earth sciences, requires an explorer to acquire relevant knowledge and perceptual skills to undertake creative problem solving and decision making. For this to be successful, an explorer must develop new hypotheses to underpin novel exploration actions. This presents an important question, regarding the role of both domain-based and broader expertise in enabling individual capacity for creative thinking, problem solving, decision making, and hypothesis generation. To gain insights into this issue, it is worthwhile examining the contemporary literature on expertise and skill acquisition.

Contemporary perspectives on skill acquisition, heavily influenced by the conceptualisation of Ecological Dynamics (ED), have resulted in the development of an understanding of expertise in decision-making at the person–environment scale of analysis [40,41,42,43]. Ecological dynamics is a multi-dimensional framework shaped by several scientific disciplines, integrated to explain human behaviours such as performance, learning and expertise, in diverse and challenging performance environments such as sport, education, and work. Major theoretical influences are provided by key concepts from ecological psychology [44], nonlinear dynamics [45] and the complexity sciences approach in neurobiology [46]. In ecological psychology, it is recognised that human behaviour is continuously regulated by information, shaping performance during activities such as the exploration of an environment [44]. Information use is based upon individual perception of affordances, which are opportunities or invitations acting to solicit or constrain behaviours within a specific performance environment [44]. The ecological approach has been enriched through the integration of tools and concepts from nonlinear dynamics, explaining how information is related to the dynamics of tasks and individuals within the performance environment. Dynamical systems theorising on human behaviour [45] propose the emergence of behavioural tendencies in perceptual, cognitive, and action sub-systems. Ecological dynamics emphasises the performer-task-environment system as the appropriate scale of analysis to explain behaviours, eschewing cognitive- or environment-biased conceptualisations of skill and expertise [47]. 

Much existing ED-related research has focused on developing an understanding of expertise, talent, and skill acquisition in sport, described by some as the most appropriate context for studying expert decision-making [48]. Ecological Dynamics takes into account the multiple dimensions of skill performance and learning, including perceptual, psychological, emotional, social, and physical aspects of the individual performer, while interacting with a specific task and environmental constraints [49]. These ideas signify the importance of the person–environment interactions at the heart of skilled behaviours, founded on the deeply integrated relationship between perception, cognitions, and actions of a performer. Based on these fundamental ideas of Ecological Dynamics, it is suggested that creative behaviours and solutions emerge during performance from continuous interactions with the environment [50,51]. This key idea implies that the emergence of creative behaviours and performance solutions is not solely a re-call of existing internalised representations or models but requires adaptation and iteration through continuous interactions with the environment, during processes of searching, exploration and discovery.

### 1.8. Perception-Action Coupling

Goal directed behaviours, such as creative problem solving, are viewed as functional coordination patterns, emerging under interacting personal, task and environmental constraints, which result in actions becoming tightly coupled to perceptual information, shaping intrinsic self-organisation tendencies in people [52,53]. Expressions of skill and expertise are continuously regulated by information. Learning is defined as the process of gradual attunement to real-time information that is meaningful, affords or supports goal-directed behavioural outcomes, and harnesses inherent system degeneracy (i.e., the same task outcomes can be achieved with different system components) [54]. Constraints, recognised as boundaries influencing behaviour, are classified into three broad categories related to the organism (the individual), task and environment [49].

Examples of mineral exploration constraints are presented in Table 1. Task constraints are aspects related to a particular goal or challenge. Although not an exhaustive list, in mineral exploration these include activity and specific goal parameters, corporate and exploration strategy statements, Key Performance Indicators (KPIs), finances and equipment. Individual constraints include the experience, attitudes and skill of individual people or teams. For example, education, values, beliefs, confidence, motivation, and risk-aversion. Environmental constraints are both physical and socio-cultural. Physical environmental constraints in mineral exploration include geology, mineralisation processes and accessibility of search areas. Company and national culture, management, reward and punishment systems, infrastructure, social networks, values, and social licence represent socio-cultural constraints that may impact the quality of decision-making. The values, attitudes and beliefs that give rise to organisation and industry culture are defined as a ‘form of life’ [55] and can significantly influence the behaviours of individuals within a system.

Within a sporting context, an example of emergent and creative decision-making might be a mid-field soccer player identifying an opportunity (affordance) to make a pass, such that an attacking player can move into a position to potentially score. Here, the mid-field player perceives unfolding opportunities to exploit space, time, and movement (constraints) as a result of representative practice experience. During mineral exploration, geoscientists are similarly required to become attuned to meaningful specifying information, as outlined in Table 1 (constraints), to predict the likely location of an undiscovered economic ore deposit that can be subjected to exploration activities (affordances) such as drill-testing or additional data collection. Hypotheses regarding the location and quality of undiscovered ore deposits are generated, tested, and adapted based on perceived affordances available to an individual, within the constraints of a specific time and situation [56]. Differences in individual knowledge, perception, motivation, and meaning shape the influence of constraints and account for diversity of perceived affordances and resultant decisions, even when identical constraints are presented to separate explorers [54]. Finally, the results of hypothesis-testing actions are observed, and the perception of constraints and affordances are subsequently adapted, both explicitly and implicitly [57]. The learning outcome is influenced by perception of action outcomes, meaning that the provision of objective and subjective feedback plays a significant role in the learning process and on-going development of expertise.

### 1.9. Learning and Expertise

Critical to improving exploration decision-making is understanding the differences between novice and expert explorers, and how one might transition from one to the other. Expertise can be loosely defined as being more attuned to specifying information, along with the creativity to recognise appropriate affordances for actions that lead to the discovery of new ore deposits. This requires the explorer to be perceptive of the constraints, including those outlined in Table 1, and able to recognise geoscientific patterns and features representative of key ore-forming processes and other specifying information within any set of dynamic and complex constraints. To do this, the explorer needs to gain experience that is representative, valid, and diverse.

For learning to be effective, it must take place in an authentic environment, containing perceptual information and affordances that are representative of the environments into which the skill will later be transferred [58,59]. If affordances are representative, then the learner can develop accurate declarative and tacit knowledge and an attunement to key information in the environment. With a broad base of relevant experiences, the learner has the opportunity to develop attunement to perceptual information in varied contexts, allowing them to recognise invariant cues and distinguish them from incidental information. This supports learners in the transfer and adaption of expertise into contexts containing novel constraints. Exploration targeting represents a complex domain, in which professionals are regularly presented with situations and tasks that are novel, requiring utilisation of a broad base of knowledge, sometimes developed for other purposes. A risk faced by mineral explorers is that their skill sets are too narrow, often focused on a single discipline (e.g., core logging, structural geology, geochemistry or geophysics), or a single deposit style or jurisdiction (e.g., orogenic gold in Western Australia, or sediment-hosted copper in the Central African Copperbelt). This represents fractionated expertise, meaning explorers are therefore not able to apply the Mineral Systems Concept to varying contexts or environments, such as the transition to new search spaces or different styles of mineralisation.

## 2. The Dynamics Model

Ericsson et al. [47] proposed that variations in individual expertise are predominantly a result of experience (e.g., “The best geologist is he [sic] who has seen the most rocks:” [60]), as opposed to factors such as innate talent. However, there are examples of experts reaching similar levels of ability within considerably different timeframes (e.g., Chess master level status reached in both 3200 and 23,000 hours of practice [61]), raising an important question regarding the influence that quality of experience has on developing expertise. 

Ecological Dynamics and other non-linear pedagogy-based research has resulted in a growing number of studies examining key factors that influence the rate at which expertise is achieved: (a) autonomy-supportive learning environments [62,63,64]; (b) motivation [65,66]; (c) effect of anxiety on performance [67]; (d) perception-action coupling [68]; (e) embodied cognition [69]; (f) affective learning design [58]; (g) development of coordinative structures [70]; (h) judgment and decision making [71], (i) the need for adaptive expertise [72]; and (j) that expertise is only gained from experience in an environment with valid cues and opportunity for feedback [73].

Based on this research, Davies and Davies [9,10] presented the Dynamics model of decision-making and learning, shown in Figure 3, to more clearly define the influence that human psychological needs and experiences have on the development of adaptive expertise. The Dynamics model is based on an Ecological Dynamics framework, specifically Newell’s [49] constraints model. The ‘energetic’ organismic (individual) constraints are separate in the Dynamics model, to highlight their importance in the design and management of learning environments. The energetic constraints combine motivation, using Self Determination Theory (SDT [34]), arousal and focus of attention. The model also includes iterative feedback loops to the energetic and more stable individual constraints, to define an autopoietic (i.e., self-organising and self-regulating) system. Key elements recognised in the model are that learning is a non-linear process [74] (Ennis, 1992) and that motivation, intention, effort and focus of attention are the initial start points, defining the quality of individual input into learning and decision-making [10,75]. Focus shifts from acquiring specific knowledge or skills toward learning-to-learn and creatively solve problems.

The model guides investigation of constraints and their complex inter-relationships. Over time, different constraints will have a greater or lesser influence in defining affordances. For example, changes in commodity prices will have a greater or lesser impact when combined with other constraints such as changes in technology, social perception, or legislation. From a skill development, creativity and risk-taking perspective, changes in constraints such as management culture, power relationships, training programs, and reward systems are all likely to have mutable and complex influences.

### Dynamics Challenge-Performance Curve

The Dynamics challenge-performance curve, shown in Figure 4, presents the complex relationship between demands or variability of a task and expected performance output, where individual, task and environmental demands, contained in the Dynamics model, are combined to represent overall challenge [76]. An important feature in the Dynamics curve is the ‘ugly-zone’ (a term coined by Alred [77]), defining the region beyond current ability in which stable performance begins to deteriorate, converging on a transition or bifurcation point. Within this zone, the learner explores solutions related to new implicit and explicit problems, providing opportunity for generation of innovative ideas, coping strategies, and step changes in understanding as new affordances. Here, the learner has an opportunity to acquire a broad and diverse range of experience, thus increasing their relative level of expertise. However, operating in the 'ugly-zone' requires confidence and resilience and can feel awkward or regressive. An integral risk of learning in the ugly-zone, is the potential for a collapse in performance, arising from the learner becoming overwhelmed or over-challenged. For this reason, it is important to recognise that, during the development of expertise, failure will likely be the *status quo*. The ugly-zone must be perceived as a place of opportunity to vary search and exploration strategies, rather than an area of risk, or failure. As such, self-determination and energetic constraints are highlighted as fundamental to creativity, problem-solving, and the development of expertise. The incorporation of ‘safe-uncertainty’ into practice environments and activities is a key environmental enabling constraint for success. 

The Dynamics curve and the Dynamics model, used together, can support the development of expertise in exploration decision making. The Dynamics model supports the identification of influencing constraints, thus providing guidance to inform the manipulation of constraints to create optimal learning experiences. Skilfully manipulating constraints will promote learners to become attentive and attuned to specifying information at appropriate levels of challenge, within representative learning environments.

## 3. Discussion

The Dynamics model and learning curve support the development of management practices and training interventions, by providing a framework for understanding and, where possible, influencing the complex interactive and adaptive constraints that shape exploration decision-making. It is evident that the complex nature of human behaviour and decision-making must be considered within the wider individual-task-environment system. This includes the impact of goal directed behaviour, carried out in a landscape of perceived affordances, within the constraints of a particular system. Training interventions that focus solely on intellectual decision-making, or the role and influence of the individual, will fail to be effective where motivational, company structure, task, socio-cultural or other limiting factors significantly shape current behaviour.

### 3.1. Constraints Shaping Exploration Decision-Making

#### 3.1.1. Goal-Directed Behaviour and Task Constraints

Goal setting is important when producing corporate and exploration strategy statements, as well as defining Key Performance Indicators (KPI’s) as guiding intentions. Strategy statements outline business-wide long-term aspirations and goals and KPI’s guide the focus of attention and the affordances perceived by employees as they work towards these goals. Well-defined goals provide clear task-related constraints for employees and teams in decision-making positions, helping to mitigate against excessive degrees of autonomy or risk-taking behaviour. Without this guidance, employees may struggle to effectively orientate their actions within an organisation, leading to splintering and ‘siloing’ of teams. Process goals are preferred over outcome goals, as they promote greater degrees of uptake and motivation [78]. Process goals define expected quality of an activity and, given the decision-maker typically has limited control over final outcomes in mineral exploration, presents an achievable objective. Realistic goals should guide, but not overly constrain, autonomous, creative decision-making within an organisation, through the provision of well-defined strategy statements and KPI’s.

The provision of strategy statements and KPIs within mineral exploration should be treated with care. Given the predominant industry focus on mineral extraction, there is a risk that those in senior management positions of a non-technical background lack a detailed understanding of the exploration process, which more closely resembles research and development activities than the mining extraction process. The degree of uncertainty associated with exploration activities often leads outcome orientated KPIs to promote non-beneficial behaviours. A simple example is the introduction of a KPI that requires an exploration team to review a large number of projects per year, leading to limited resources being stretched across those projects, thus reducing the quality of each individual review. 

#### 3.1.2. Company Culture, Leadership, and Other Environmental Constraints

Several efforts have been made to define key aspects of company culture that influence exploration success. Towards developing a philosophy of oil exploration, Wallace Pratt [79] commented that “oil is first found… in the minds of men,” recognising the value of vision and creative thinking in exploration. Masters [80] suggested that adopting the characteristics of a small company was vital to success in oil exploration, trading control and routine for innovation, motivation, and speed, passing power down and providing employees with the freedom to both grow and fail.

Most notable within the minerals industry are the writings and interviews of Roy Woodall [6], widely recognised as playing a significant role in the success of Western Mining Corporation (WMC). Interviewed by Stanton [81], Roy Woodall stated that building a successful exploration company starts with the people, and that he conducted recruitment personally. The successful WMC team was brimming with creative intellectual energy but could be a tough group of individuals to manage. In their report on the successful management of minerals exploration, McKinsey Company [82] stated that management and leadership play an important role in exploration success, and that “good explorers can be made as well as born.” In an SEG Newsletter, Dan Wood [83] considered that exploration is “an art informed by science.” Amongst other things, Wood suggested that risk-taking, creativity and a ‘discovery culture’ are critical areas that govern exploration success, sometimes preferring low-tech solutions that focus creative thinking over the latest computational or statistical methods. Wood and Hedenquist [84] suggest that improvements to business models and the predictive way geologists think when exploring are key to improving exploration success rates, with technological developments playing only a supporting role.

Although many companies within the minerals industry aspire to creating learning organisations, there is a persistent culture of error and risk aversion, lack of feedback, unfilled ‘near-miss’ books, and of rewarding ‘safe’ decision-making. Decision-makers must develop the ability to conduct appropriate risk-benefit analyses, where risks are identified up-front and considered against their potential exploitation, rather than simple elimination or justification [71]. Company reward and punishment systems [85], as well as an ego element of not being seen to make mistakes [72], further constrains creative decision-making and learning within mineral exploration. Critical to improving culture within the minerals industry is the degree of autonomy provided to each exploration geologist and an acceptance of the uncertainty and technical risk associated with conducting exploration and learning within the ugly-zone. However, providing autonomy, as part of a needs-supportive environment, should not lead to abdication of responsibility by management.

#### 3.1.3. Motivation, Autonomy, and Risk-Taking

In his 2012 paper, Andrew Curtis [86] stated that “scientists should… not be ashamed of subjectivity, but should strive to… reduce its effects.” This statement stands contrary to most research conducted into human intuition and decision-making, where decisions are made within a set of highly complex constraints, with incomplete knowledge of a context and expected outcome. Decisions, such as choice of research questions, methodologies, and interpretation of results, are fundamental to the scientific method, but are rarely measurable or quantifiable prior to their completion. As such, it is advised that exploration geologists embrace subjectivity in decision-making, with recognition of the importance of experience and creativity in making successful subjective estimates and interpretations, especially in conducting predictive exploration targeting.

As outlined by the Dynamics challenge-performance curve (Figure 4), the ugly-zone should be perceived as a place of opportunity. By increasing variability, there is opportunity to become attuned to new affordances, ultimately driving effective learning and the development of more adaptive expertise. Many of the constraints presented within mineral exploration are project-specific, meaning the exploration geoscientists operating ‘on the ground’ have access to the most detailed and up-to-date information. As such, the responsibility for decision-making will typically be in collaboration with more junior employees. To support these decision-makers, management must clearly articulate corporate strategy, goals, and broad environmental constraints.

Additionally, attaining a culture of creativity and adaptive expertise requires the motivational climate of a company to support the psychological needs of the employees, providing an appropriate degree of autonomy to take risks, without undermining their ability to learn or perform. In any context, humans are inherently driven towards personal development and satisfaction of three basic psychological needs: autonomy, competence, and relatedness [87]. If psychological needs are met, humans tend to possess greater motivation and feelings of competence [88], actively seeking out meaning and cues to support autonomous decision-making [34]. In contrast, when tired, under pressure, or emotionally challenged, humans are susceptible to heuristic biases, substitution, and risk-aversion [89]. 

#### 3.1.4. Practical Training Interventions

Given shareholder capital is generally used to conduct exploration activities, there is a requirement for exploration companies to mitigate against excessive risk-taking. In this instance, the operating constraints within the industry reduce the potential of the learner to develop adaptive expertise, by limiting their ability to test and adapt innovative hypotheses or attune to affordances by exposing themselves to challenging decision constraints. Davies et al. [39] also recognised that certain aspects of exploration targeting are likely to have low-validity [89], where noisy or highly complex situations present a lack of decision-feedback, rendering genuine expertise unachievable through typical work-related experience and learning. This is compounded by the limited number of projects upon which an exploration geologist is likely to have worked during their career.

Although several studies have highlighted the ability of algorithms to outperform professionals in low-validity environments [90], these studies found that forecasts made by algorithms were generally incorrect, albeit less often than in human predictions. The superior performance of algorithms in low-validity environments is attributed to consistency [91], an undesirable trait in mineral exploration, where targeting activities are in competition with previous explorers, signifying that consistent or repeated activities are more likely to result in failure after an initial effort (sampling without replacement). To mitigate against these limitations, a number of on-the-job training methods have been proposed, including: observation of experts, professional discussion in communities of practice, experimentation with differing strategies, engagement in after-action reviews, and coaching from others with wider experience [92]. However, since certain aspects of exploration targeting are recognised as challenging and potentially low-validity, it is unlikely that sufficient expertise exists within industry to conduct wide-spread, on-the-job training [39]. 

Hogarth and Soyer [93] and Singer [94] suggest that simulated experiences can be provided to promote the development of expert intuition. As such, it is advised that practical scenario-based exploration targeting training courses should be developed, perhaps using virtual or augmented reality environments, based on real scenarios to ensure that contextual information is relevant and valid. Using these technologies, training environments can be designed appropriately for each individual learner, providing a representative learning environment within which decision constraints are carefully selected to promote positive exploration behaviours, as well as the recognition and understanding of key explicit and implicit information in a realistic exploration environment. This is achieved through highlighting key information sources, providing appropriate feedback, and promoting development of perception-action coupling, expertise, and resilience within a needs-supportive learning environment.

Such training interventions could involve presenting the learner with information (constraints) relating to the location of economic ore bodies. Using this information, the learner would present their hypotheses regarding the location of ore bodies and propose exploration programs (based on perceived affordances) to test these hypotheses. The complexity of decision constraints would be adapted based on the level of expertise of each participant. The design of these courses would utilize simulation programs, presenting the learner with realistic scenarios containing real exploration data to match future performance environments [10,46] and include sources of both implicit and explicit knowledge [57]. To achieve this, participants would be provided with comprehensive geoscientific datasets, but characteristics and locations of known deposits would be withheld. Each individual participant would work through the exercise to produce predictions for the size, quality, and locations of undiscovered deposits within the study area, and devise an appropriate exploration program with which to test these hypotheses. The training course methodology would outline a broad framework, but still allow a high degree of individual flexibility. This allows each participant to adapt individual strategy to gain maximum benefit from their personal background and expertise, as well as devise novel affordances for testing varied hypotheses. Although initial predictions would be conducted in isolation, to promote engagement and divergent thinking [78], the exploration team could then come together to discuss individual predictions and reasoning, allowing for discussion and a group consensus to develop. Finally, upon completion of the exercise for a given study area, known deposit information would be presented to the participants to simulate the results of a comprehensive exploration program. This provides immediate feedback to the participants, allowing them to compare their predictions with the real-world deposit information. Opportunities for feedback, comparison and reflective learning might also be improved through guidance from qualified trainers. By conducting such training exercises, for a suite of study areas covering different geological environments, deposit styles, and jurisdictions, it would be possible for an exploration team to acquire experience, typically gained over decades in an exploration career, in a matter of days or weeks. Most importantly, this experience would be acquired in a safe environment, without risk to either career progression or shareholder capital. 

## 4. Conclusions

It is proposed that an increase in creative and challenging problem-solving exercises, during application of the Mineral Systems Concept, has the potential to result in the development of widespread search expertise and improved long-term exploration outcomes. Through application of the Dynamics model, influential constraints can be identified and leveraged. Constraints that are limiting creative problem solving, for example traditional risk-adverse company culture, can be reduced, while enabling constraints can be amplified. In combination with the Dynamics challenge-performance curve, several areas are recognised as having significant influence over the quality and creative aspect of exploration decision-making. Explorers require a degree of autonomy to be confident enough to enter the ugly zone, take risks, and test novel search ideas. Clear goal-oriented strategies provide a focus of attention to relevant information, and management can provide a framework to mitigate and challenge excessive risk-taking and minimise consequences of inevitable failures. Finally, development of appropriate scenario-based training courses is identified as a critical suggestion, providing an opportunity for explorers to develop an accurate perception of situational constraints, and conceive affordances for testing novel exploration targeting hypotheses, thus significantly improving the quality of decision-making and learning outcomes throughout the exploration industry. Since energetic individual constraints define the quality of individual input into learning and decision-making tasks, each of the above suggestions must be provided within needs-supportive learning and working environments so as to maintain motivation, effort, and focus of attention throughout the exploration process.

Adoption of these concepts is required by the industry to present working environments that allow for effective application, and further development, of creative exploration targeting expertise. Further research is suggested, regarding each sub-discipline and their application to mineral exploration, as well as other industries and research organisations.

## Figures and Tables

**Figure 1 ijerph-18-09752-f001:**
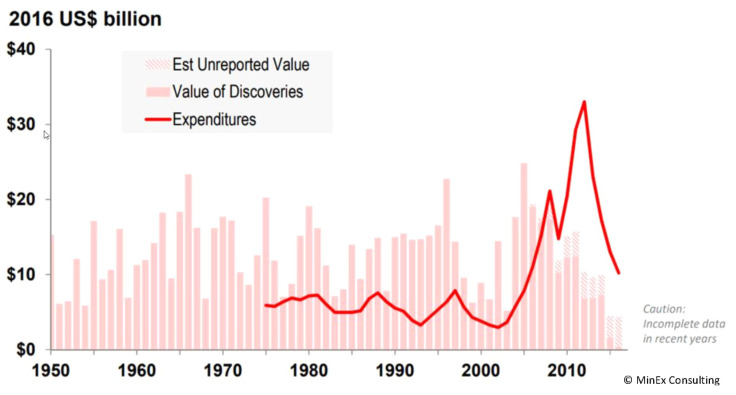
Declining rate of exploration success during last decade, despite increased exploration expenditure since mid-2000’s. Adapted with permission from Schodde, R.C [1].

**Figure 2 ijerph-18-09752-f002:**
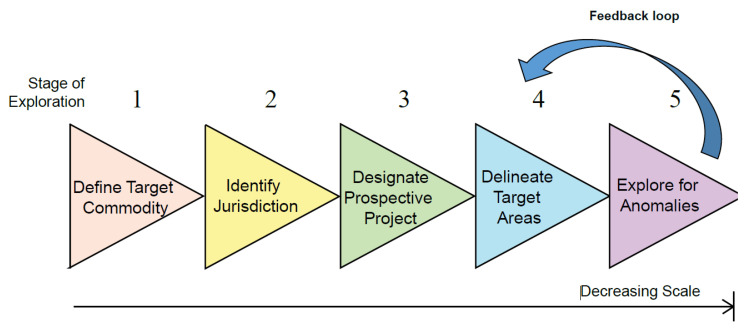
The stages of exploration targeting. Within each stage there is a period of data collection, integration, and analysis. Moving between stages requires a reduction in real options, such that a decision is made to focus on a specific area, decreasing the overall exploration search space. Although a single feedback loop is presented, it is noted that the true process of exploration is far less rigid. Feedback occurs at all stages, potentially leading to a reverse step in the process. For example, newly acquired data may present a case for defining a different, or second, target commodity within a project area.

**Figure 3 ijerph-18-09752-f003:**
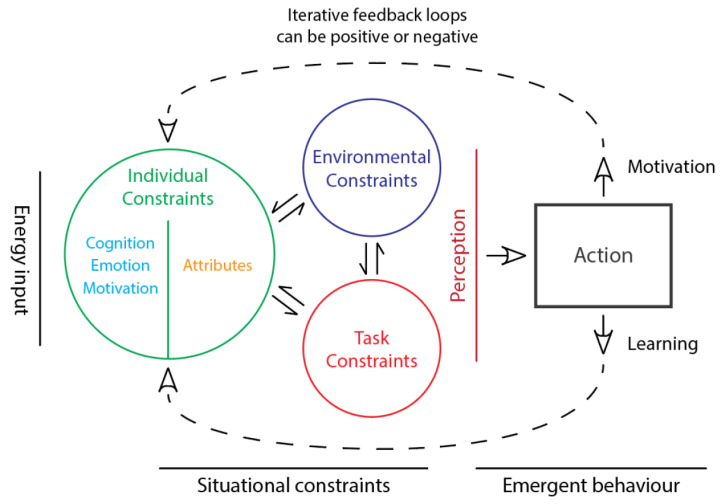
Dynamics model for decision-making and learning, adapted with permission from Davies, M.J. and Davies, R.S. [9]. Motivation is the initial start point, defining the quality of individual input into learning and decision-making tasks. Decision-making behaviours are shaped by intentions and perception of available affordances, within situational constraints. These constraints are broadly categorised into individual, environment, and task. Differences in individual perception, motivation and meaning shape the influence of constraints and account for diversity of decisions. Within mineral exploration, these decisions can include activities such as collecting additional geoscientific data (e.g., airborne geophysical surveys, surficial geochemical sampling), direct drill-testing of targets, or even choosing to relinquish ground. The results of hypothesis-testing actions are observed, changes in perception and decision-making may be implicit or explicit making them more difficult to identify, the influence of constraints may be redefined, and hypotheses subsequently adapted. This process of active learning results in perception-action coupling, where feedback or results have an impact on individual learning and motivation, through a learning feedback loop, defining an autopoietic (i.e., self-organising and self-regulating) system.

**Figure 4 ijerph-18-09752-f004:**
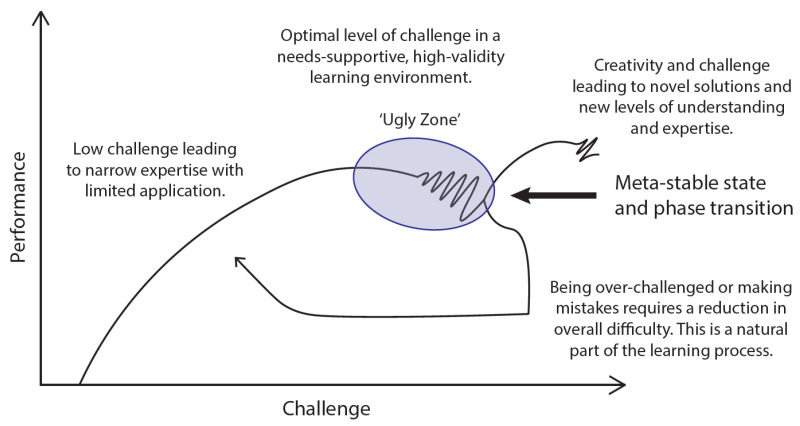
Dynamics challenge-performance curve, adapted with permission from Davies, M.J. and Davies, R.S. [10]. The complex relationship between demands of a task and expected performance output are represented by a performance curve, where individual, task, and environmental demands are combined to represent overall challenge. Limited overall demands result in a performer remaining under-challenged, lacking the opportunity to expand their expertise. With increasing difficulty, performance trends upwards towards the ugly-zone; a point of optimal challenge, where the opportunity to develop expertise is maximised, as new implicit and explicit affordances become available through interaction with dynamic constraints in real-time. Within the ugly-zone, performance becomes unstable, converging on a bifurcation point and presenting the risk of errors being made, or a drop in performance arising from the learner becoming overwhelmed or over-challenged. The curve shows that to get back to the optimal performance, overall challenge needs to be reduced significantly to allow reflection and learning (hysteresis). Despite this risk, it is important that the ugly-zone be perceived as a place of opportunity for learning and development of expertise within a well-managed, needs-supportive learning and performance environment.

**Table 1 ijerph-18-09752-t001:** Examples of constraints in mineral exploration targeting.

Group	Constraint	Description
Environmental	Geology	Geology of an exploration project (much of this remains unknown, as only the geoscientific datasets outlined under task constraints are available to the explorer)
	Company/culture	Organisational structure, explicit and implicit rules or values
	Management/leadership	Methods and styles applied to guiding individuals and teams within an organisation
	Government/mining law	Political landscape and specific laws governing mineral exploration/extraction
	Social perception	Social landscape and social licence to operate
	Land access	Access to exploration ground, defined by stakeholders, law and availability
	Market	Factors influencing commodity price and availability of investment funding
	Academia	Academic institutions conducting research and training students
	Economic geology theory	Current level of geoscientific theory and knowledge relevant to mineral exploration
	Exploration and mineral processing technology	Current technology available to the minerals industry
Individual/team	Individual attributes/skills	Knowledge and expertise of individual geoscientist
	Individual attitudes/motivation	Individual psychology, including motivation, attitude, values
	Team structure	Selection of individuals with complementary skills, working in a collaborative manner
	Team capabilities	Capabilities of team due to individual skills, psychology, teamwork, and corporate resources
Task	Geoscientific datasets	Geoscientific datasets that relate to the location of undiscovered ore deposits (geological, geochemical, geophysical, geochronological, etc.)
	Corporate/exploration strategy	High-level strategic plan and goals (company vision)
	Key Performance Indicators (KPIs)	Measurable performance outcome specific to a particular activity
	Finances	Availability of finances to conduct exploration activities
	Service providers/equipment	Consultants, research institutes, technology providers, and in-house company equipment
	Training/CPD	Training and professional development

Those highlighted in grey are typically recognised as key constraints in mineral exploration, although the remaining constraints outlined here also play a significant role in influencing decision-making.

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
