# Peer review of "Learning and Expertise in Mineral Exploration Decision-Making: An Ecological Dynamics Perspective"

_ijerph, 2021, doi:10.3390/ijerph18189752_

Round 1

Reviewer 1 Report

The review paper Learning and Expertise in Mineral Exploration Decision-making: An Ecological Dynamics Perspective approaches an interesting issue of Mineral Exploration: what is the best model for discovering new mineral reserves?

Before reading the text, and based on the affiliations of the authors, I presumed this was a multidisciplinary paper. And I am right! It was a great pleasure to read the paper and to learn a lot with it.

I hope I can help with some comments in order to improve this great work.

First, I would like to see a better description about the ‘creative thinking’ interpretation of the authors. What is creative thinking in mineral industry? Is it finding new answers or looking for new questions?

Another point that was a little abstract for me was the Ecological dynamics. The concepts are brillinantly presented, but what is the relation of Ecological dynamics and the Mineral Exploration? In the other itens, the authors contextualized the concept and the Mineral Exploration application. I suggest the same approach method here.

The two graphic images are objective, clear and very general, I mean, they approach a method applicable to any problem. So, as the paper is about a method AND about Exploration, I would like to suggest that you insert an image into the conclusions or discussions contextualizing this application of the method in exploration.

Finally, I believe that some typing problem caused item 1.5 title to repeat itself, so it needs to be revised.

Once again, thanks for the opportunity to review this great work. I look forward to seeing it published soon.

best regards,

Reviewer 2 Report

The article is a comprehensive interdisciplinary study of the problems of the decision-making process in the planning and exploration of mineral deposits. The authors have conducted a very comprehensive literature review covering the human sciences, psychology, management theory, economics, finance. The authors demonstrate high professionalism, erudition, ability to holistically approach the problem of mineral exploration. They refer to their previous works concerning decision making process in mineral deposits planning and prospecting. The authors describe and recommend the use of the Mineral Systems Concept in the decision-making process. They present conditions and benefits of using this system (Mineral Systems Concept). They draw attention to the need for autonomy and independence of experts in the exploration process. Much space is devoted to risk assessment in decision making. 

This so important text ends with rather vague conclusions. In my opinion this is the only weakness of the text. It seems necessary for the Authors to make the conclusions a bit more detailed and specific. It is especially about recommendations for mineral companies and their top management.

I have no  editorial comments on the text.

Reviewer 3 Report

This manuscript does an interesting demonstration of an ecological dynamics perspective learning and expertise in mineral exploration decision-making. Introduction is very good, however suggested inclusion of additional information and modifications will definitely improve the standard and quality of the article.

The authors should explain what new and original this paper has to offer beyond the already existing in the literature. What makes this work different from others? It should be better explained.

Abstract:

‘’Here, we outline the Dynamics model, as a diagnostic tool for situational analysis and a guiding framework for designing working and training environments to maximise exploration performance.’’

If the Dynamics model want to be used as a diagnostic tool for situational analysis, then we need some data and validation, any more information in this regard?

Formatting: It will be great if all headline have the same format and follow the Journal formatting standards.

  1. Introduction

2 Literature Review

3 The Dynamics Model

Section 1 and 2 could be combined as normally Literature Review covered in  Introduction  section.

Unfortunately, after of all of these remarks I must suggest that this manuscript should be rewritten and resubmitted to the Journal, because I find that it has not the sufficient quality to be published in present state.

Round 2

Reviewer 3 Report

Unfortunately, after of all of these remarks I must suggest that this manuscript should be rewritten and resubmitted to the Journal, because I find that it has not the sufficient quality to be published in present state.